# Relationship of Prevalent Fragility Fracture in Dementia Patients: Three Years Follow up Study

**DOI:** 10.3390/geriatrics5040099

**Published:** 2020-11-30

**Authors:** Inderpal Singh, Daniel Duric, Alfe Motoc, Chris Edwards, Anser Anwar

**Affiliations:** 1Department of Geriatric Medicine, Ysbyty Ystrad Fawr, Aneurin Bevan University Health Board, Ystrad Mynach, Wales CF82 7EP, UK; alfe.motoc@wales.nhs.uk (A.M.); anser.anwar@wales.nhs.uk (A.A.); 2Health Education and Improvement Wales (HEIW), Caerphilly, Wales CF15 7QQ, UK; daniel.duric@wales.nhs.uk; 3Research and Development Department, Aneurin Bevan University Health Board, Newport, Wales NP20 2UB, UK; chris.edwards3@wales.nhs.uk

**Keywords:** fragility fracture, hip fracture, dementia, mortality, osteoporosis

## Abstract

**Introduction:** dementia increases the risk of falls by 2–3 times and cognitively impaired patients are three times more likely to have hip fracture following a fall when compared to cognitively intact individuals. However, there is not enough evidence that explores the relationship between dementia and fragility fractures. The aim of this study is to explore the relationships of prevalent fragility fracture in patients with dementia admitted with an acute illness to the hospital. **Methods:** the existing Health Board records were reviewed retrospectively for all patients admitted diagnosed with dementia in the year 2016. All patients were followed up for a maximum of three years. All of the the dementia patients were divided into three groups: group 1—“no fractures”; group 2—“all fractures”; group 3—“fragility fractures”. Clinical outcomes were analysed for hospital stay, discharge destination (new care home), post-discharge hip fracture data, and mortality. **Results:** dementia patients with a prevalent fracture were significantly older, 62% were women. A significantly higher proportion of dementia patients with prevalent fractures were care home residents and taking a significantly higher number of medications. The mean Charlson comorbidity index was similar in patients with or without fracture. Dementia patients with a prevalent fracture required a new care home and this is significantly higher when compared to those with no fracture. Mortality at one year and three year was not statistically different in patients with or without prevalent fractures. A significantly higher number (21.5%) of dementia patients with prevalent fragility fracture sustained a new hip fracture when compared to those with no prevalent osteoporotic fracture (2.9%) over the three years follow up (*p* < 0.0001). **Conclusion:** dementia patients with a prevalent fragility fracture is associated with a statistically significant higher risk of a new care home placement following acute hospital admission. This sub-group is also at risk of a new hip fracture in the next three years. Whilst clinical judgement remains crucial in the care of frail older people, it is prudent to consider medical management of osteoporosis in dementia if deemed to be beneficial following the comprehensive geriatric assessment.

## 1. Introduction

Worldwide, the count of people diagnosed with dementia is projected to be increased from 50 million in 2018 to 152 million in 2050, which is over a 200 percent increase [1]. Dementia UK (2014) reported a similar exponential increase in the number people with dementia in the United Kingdom (currently 850K) to one million by 2021 and it is expected to be over two million by 2051. The dementia care in the United Kingdom is costing 26 billion pounds a year [2]. Approximately one-third of the unplanned medical beds in the UK are occupied by dementia patients; therefore, delivering high quality dignified care is high on the health service agenda [3,4].

Dementia patients could also have other associated long-term conditions; therefore, they are at a higher risk of adverse clinical outcomes [5,6]. Dementia has been largely associated with hip fractures in the community and increases the falls risk by 2–3 times [7] and studies have reported a significantly higher risk of inpatient falls in dementia patients as compared to those with intact cognition [8]. Dementia has been reported as the most common comorbidity (63%) that is associated with inpatient hip fractures (IHF) [9].

The risk of dementia and hip fractures increases exponentially with age [10,11]. Both dementia and hip fracture are associated with adverse clinical outcomes, financial loss, and they have an impact on the quality of life on both patients and caregivers [12,13,14,15]. On occasions, the presence of dementia could be used as justification for therapeutic nihilism, which makes them more vulnerable as compared to a cognitively intact older adult [15,16]. Prudent healthcare principles have been commissioned in Wales to improve quality of care, targeting those individuals with maximum health care need [17]. Older people with dementia and previous falls and osteoporotic fracture, particularly hip fracture, are the frailest. Based on the quality criteria set by the Bevan Commission, osteoporosis treatment should be offered if appropriate and based on the existing resources and current evidence [17].

The objective of this study is to measure the impact of prevalent fractures (all fractures and fragility fractures) on clinical outcomes and the incidence of post-discharge hip fracture in patients with dementia admitted acutely across three acute care hospitals within Aneurin Bevan University Health Board (ABUHB). This would help us to explore quality improvement initiatives in order to improve patient care and plan our future services accordingly.

## 2. Methodology

### 2.1. Design

The study was retrospectively designed in order to profile the clinical outcomes of acutely unwell older people with known dementia admitted to the medical admissions units.

### 2.2. Setting

Dementia patients presenting either of the three enhanced general care hospitals within one Health Board in Wales from 1 January 2016 to 31 December 2016 were included.

### 2.3. Data and Statistical Analysis

Information on age, gender, residence, long term conditions, drugs, and any fracture before index admission was recorded in an Excel spread sheet from the Health Board records and various clinical letter communications between primary and secondary care. The Charlson Comorbidity index (CCI) was calculated based on the listed co-morbidities in the discharge summaries and GP referral letters.

Fragility fractures are defined as fractures that result from a very low energy trauma, such as a fall from a sitting or standing height [18]. The most commonly observed fragility fractures include the spine (vertebrae), the proximal femur (hip), distal radius (wrist), pelvis, and humerus [18]. In this study, fragility fractures and any other fracture, including ankle, ribs, tibia, and fibula, were classed as all fractures. All the dementia patients were divided into three groups: group 1—“no fractures”; group 2—“all fractures”; group 3—“fragility fractures”, and were reviewed following discharge for a maximum of three years until the 31st December 2019. The appropriate control group would be “group 1—dementia patients with no fractures”.

Prevalent fragility fractures were identified from previous X-rays and any fracture reported in the previous five years were included. Prevalent fractures were not differentiated any further by the number of fractures. Clinical outcomes were analysed for hospital stay, discharge destination, post-discharge hip fracture data, and mortality. In this study, care home refers to both residential and nursing home.

The results are presented as means along with standard deviation. The *t*-tests were used in order to compare the means, and chi-square tests were used to compare categorical variables. The mortality results among three groups were analysed while using the difference between proportions test. The association between history of fracture and mortality was evaluated through Cox regression. In this study “poor outcome” was defined as either in-patient mortality or discharge to a new care home and analysis were completed while using binary logistic regression for factors, including age, gender, comorbidities, drugs, and residence at time of admission. The risk estimates for the chance of presenting with a new hip fracture or care home admission were done while using multiple regression with binomial distribution and logit link function for factors influencing the risk of a hip fracture during admission and after discharge.

The STATISTICA StatSoft data analysis software system, version 9.1 (StatSoft, Inc., 2010, Tulsa, OK, USA) was used. Because we are comparing two groups to the same baseline, this is multiple comparisons, which means that *p* = 0.05 is not an appropriate threshold to accept statistical significance, since two comparisons, each at 5% probability of being wrong, raise the probability of a false significance. To overcome this, we have divided the acceptable significance level by the number of comparisons. Thus, in this study, we have set the significance level at *p* = 0.025.

This study was constituted as an evaluation of the existing service where patients are routinely admitted for their health care needs. In view of the retrospective and non-interventional nature, this study did not meet the requirement for any further ethical approval process. This study was discussed with Research and Development department of the Health Board and it was deemed that patients cannot be identified and will not be contacted as part of the evaluation; internal approval was granted to review the existing patient data. Anyone involved in data collection or data analyses followed general data protection regulations.

## 3. Results

### 3.1. Demographic Profile

A total of 2067 dementia patients that were admitted to an acute care hospital were included in this study. Fifty-seven patients were excluded due to missing data. The mean age of all patients included was 84.5 ± 7.7 years, 62.3% were females.

In the control group (group 1, *n* = 1363), the mean age was 83.8 ± 8.0 years, and 55% female (752/1363). The mean number of comorbidities was 5.0 ± 2.0, mean CCI 5.9 ± 1.5, mean number of drugs 7.8 ± 3.8, and mean length of stay after first admission 17.4 ± 23.4 days.

The mean age of dementia patients with prevalent fracture was 85.9 ± 6.9 years. This was significantly higher when compared to those without prevalent fractures (83.8 ± 8.0, *p* ≤ 000.1). More than 75% of dementia patients who had prevalent fractures were females.

Seventy-three percent of dementia patients with no prevalent fractures were living in their own homes. In comparison, a lower proportion of dementia patients with prevalent fractures were living in their own homes (66.8%, *p* ≤ 0.0001) and this was statistically significant (Table 1). Conversely, a higher number of dementia patients (30.9%) with prevalent fractures were living in care homes as compared to those without prevalent fractures (24.7 %), which was, again, a statistically significant difference, *p* = 0.003) (Figure 1).

### 3.2. Clinical Characteristics

Although patients with prevalent fractures were prescribed significantly more medications, there was no statistically significant difference in the CCI in the three groups. Table 1 shows the detailed profile and clinical characteristics of all dementia patients for the three groups (no fracture history, all fractures and fragility fractures).

### 3.3. Clinical Outcomes

Table 2 shows the clinical outcome data of all dementia patients for three groups (no fracture history, all fractures and fragility fractures). The mean LoS for all dementia patients was 17.8 ± 23.4 days and there was no statistically significant difference in people with dementia with or without prevalent fragility fractures (Table 2).

### 3.4. Discharge Destination

Only 30% (*n* = 183) dementia patients with a prevalent fracture were discharged back to their own homes, which is less than half of the total (*n* = 396) admitted from own homes. In comparison, more patients with pre-existing dementia, but without prevalent fracture (46.3%), were discharged to their own homes and this was statistically significant (*p* ≤ 0.0001) (Table 2). Conversely, more patients with underlying dementia with a prevalent fragility fracture (52.6%) required a new care home following an acute admission as compared to those without prevalent fractures (32.2%, *p* = 0.0001) (Figure 2). 

Overall, 31.4% dementia patients with prevalent fragility fractures required a new care home as compared to those without any fracture (15.5%), which is more than double and also statistically significant, *p* ≤ 0.0001) (Figure 3).

### 3.5. Mortality

The overall inpatient mortality and one-year mortality rates were 15.6 % (*n* = 324/2067) and 48.6% (*n* = 1005/2067). There was no significant difference in the in-patient, 30-days, or one-year mortality in these three groups (Table 2). There was no significant difference in three-year mortality between the no fracture or fragility fracture group (*p* = 0.03). The mortality trends between no fracture and fragility fracture group has been shown using Kaplan–Meier survival curves and there is no significant mortality difference in these two groups (*p* = 0.54) (Figure 4).

### 3.6. Poor Outcome Predictors

Sub-analysis on the basis of Cox regression indicates age on admission is significant risk factor in mortality (Hazard Ratio (HR) 1.04 (95% CI = 1.032–1.048), *p* < 0.0001). Fractures do not pose significant risk (all fractures—HR 1.006 (95% CI = 0.901–1.123); fragility fracture HR 0.993 (95% CI = 0.886–1.113); and, hip fracture during index admission and after discharge HR 1.013 (95% CI 0.768–1.336).

Using multiple regression with binomial distribution and logit link function, we tested for factors influencing the probability of a hip fracture after discharge from the hospital. Significant factors were age on admission (*p* = 0.033), gender (*p* = 0.047), prevalent all fractures (*p* < 0.0001), and prevalent fragility fracture (*p* = 0.0002).

The two primary factors with a significantly higher risk of poor outcome (either in-patient mortality or discharge to a new care home) using binary logistic regression were age on admission (*p* = 0.00015) and CCI (*p* = 0.021).

### 3.7. Follow-Up Hip Fracture

Overall, the post-discharge hip fracture rate over three years for 1743 patients (*n* = 2067−324) was 10.3% (*n* = 155/1743). More than one-fifth (21.5%) dementia patients with prevalent fragility fracture sustained a new hip fracture when compared to those without prevalent fracture (2.9%) over the three years follow up. This was significantly higher (*p* < 000.1). Sixty patients (2.4%, 60/2067) sustained an IHF. The rate of IHF was higher for those dementia patients who had prevalent fragility fracture, but this was not a statistically significant difference (*p* = 0.04) based on an appropriate threshold to accept statistical significance for this evaluation.

## 4. Discussion

Dementia is a progressive disorder of cognition and it results in 2–3 times higher fall risk [7]. Cognitive impairment and fragility fractures are common, and both could be invariably present in older people. A higher rate of fragility fracture, including hip fracture, has been observed in people living with dementia when compared to cognitively intact individuals [19,20,21,22]. Similarly, osteoporosis has been observed as an early risk factor for dementia [23,24]. Comparatively higher mortality and a higher discharge rate to a new care home for this cohort of the most vulnerable population are self-explanatory. However, there is not enough evidence to suggest a very clear relationship between fragility fracture and dementia.

Ageing has been associated with increased prevalence of dementia and overall dementia prevalence is increasing globally. However, most recent secular trends in the diagnosis of dementia have shown a decline in some high-income Western countries in the last two decades [25]. Similarly, the incidence of osteoporosis and fragility fracture increases with the longevity of the population, but age is only one of the several risk factors, including reduced bone density, the use of glucocorticoids, gender, prevalent fractures, and family history of osteoporosis, genetic or environmental factors, endocrine abnormalities, and malabsorption syndromes [26].

Age could be accepted as a joint risk that is associated with both dementia and hip fracture [27], but there is not enough data to support this. In this study, we observed that the mean age was significantly higher in dementia patients with prevalent fracture (85.9 ± 6.9) years as compared to those with only dementia and no prevalent fragility fracture (83.8 ± 8.0). In this study, higher age on admission was also associated with risk of new hip fracture following discharge or a need for a new care home placement. It was also observed that more than 75% of dementia patients who had prevalent fractures were females.

Nursing home residents have shown higher mortality, particularly those with advanced cognitive impairment [28]. Overall, about one in ten people with a hip fracture could die within one month and another one-third could die within 12 months [28]. In this study, the overall mortality in patients with dementia with prevalent fracture at 30 days and one year was 20% and 50%, respectively, which suggested that dementia is associated with poorer outcomes.

In the UK, 30% community hip fractures are admitted to hospital from residential or care homes [29]. In this study, 27% of dementia patients were admitted from a care home. Half of the dementia patients with prevalent fragility fracture required a new care home as compared to a third in those without prevalent fracture, which was significantly lower and, therefore, supports fracture risk assessments for dementia patients in the care homes.

A substantial treatment gap has been reported in the majority of individuals living in the European Union who had previous fragility fracture or are at a risk of osteoporotic fracture. [30]. Despite the high osteoporosis health care burden and economic cost of fragility fractures, the recommended use of anti-resorptive drugs has decreased in recent years [30].

There is enough evidence in the literature that any prevalent fragility fracture exponentially increases the risk factor for subsequent fractures [31,32,33,34,35]. The highest probability of having a new fracture is within two years, but risk is highest immediately following the initial fracture [34]. This risk is similar in both men and women and it could persist for 10 years [35]. This study highlights an unmet need for bone health assessment in dementia patients. Prior fragility fracture does indicate reduced bone mineral density; therefore, a higher risk of future fracture. This study specifically examined acute patients with dementia and comparing outcomes in those with and without a prior fracture. Therefore, it is prudent for dementia patients to have osteoporosis risk assessment and be treated with a combination of vitamin D, calcium, and/or bisphosphonates if appropriate and deemed at fracture risk.

We acknowledge the weaknesses of this study. The burden of fragility fractures in people with dementia admitted acutely to hospital is measured, retrospectively. The total number of drugs were counted from the discharge summaries to see the impact of polypharmacy. The class of the drug, details of the dementia drugs, anti-psychotics, or anti-resorptive drugs was not available for all patients as General Practitioners (GP) records, therefore were not accessed as part of this study.

Although the mean CCI did not show any significance among the three groups, further detailed analyses for different chronic conditions with fragility fractures or clinical outcomes was not undertaken. This limits the generalisability of the prevalent fractures in the primary care and community. The analyses are based on sample from one Health Board and data were only retrieved from the existing information. Therefore, patients that were treated for fractures in other Health Boards or Trusts might have been missed in all the three sub-groups. The research team had tried to overcome this weakness by reviewing GP referral letters that have the most updated clinical information. Selection bias was another weakness of this study and, to minimise this bias, all of the patients with diagnosis of dementia were included. The data on severity of dementia were not available on discharge summary; therefore, the impact of severity of dementia was not studied on poor outcomes.

This study has been strengthened by the large sample size and long follow up of acute dementia patients up to three years for subsequent hip fracture. Prevalent fractures in the last five years were validated by reviewing reported X-rays. In this study, the overall, post-discharge hip fracture rate over three years for 1743 patients was 10.3%. A significantly higher proportion of dementia patients with prevalent fragility fracture (21.5%) sustained a new hip fracture when compared to those dementia patients with no prevalent fracture (2.9%) over the three years follow up. The authors are not aware of any other study on dementia patients to report hip fracture for three years follow up following an acute admission.

The current guidance recommends early diagnosis and management of dementia. Similarly, guidance has been focused on the primary and secondary prevention of osteoporotic fractures to improve clinical outcomes and reduce healthcare cost and socioeconomic burden [26,36,37]. A combination of lifestyle changes and drug treatment has been proposed in order to treat patients with osteoporosis [38]. It is challenging to prevent falls, more so in patients with advanced dementia, but multifactorial assessment for potentially modifiable intrinsic and extrinsic risk factors could reduce falls in patients with early dementia.

It is well-known that individuals in the general population with prevalent fractures are exposed to a higher risk of a new fracture event. Calcium has been in discussion over centuries to strengthen bones, thus preventing fractures by calcium supplementation in toddlers to very frail care home residents. The rate of age-related bone loss both in men and women following middle-age is approximately 0.5–1.0% per year [39]. Some studies suggest with appropriate calcium supplement with or without vitamin D can prevent bone loss, possibly due to the effect of calcium in suppressing PTH secretion [40,41]. However, most observational studies do not show a relationship between calcium intake and fracture risk and meta-analysis of randomized clinical trials does not support the routine use of calcium, vitamin D, or both in reducing risk of fractures among community-dwelling older adults [42]. Over the last decade, experimental studies, calcium metabolism, and dysregulation of neuronal calcium signalling and alterations in the expression of calcium signalling proteins have been observed in animal models of familial Alzheimer’s disease (AD) [43,44,45] and this a possibility of correcting neuronal calcium disruption as a therapeutic approach for AD [45]. There be could be a role of leptins in osteoporosis [46], inositols [47], or Calmodulin [48] for delaying cognitive impairment, but this needs further research. The current recommendation suggests calcium and vitamin D supplementation in those receiving treatment for osteoporosis and for care home residents who are at high risk of calcium and vitamin D insufficiency [49].

This study specifically adds to existing literature that dementia patients are more likely to have poorer outcomes; therefore, dementia should not be used as justification for therapeutic nihilism. This study also supports the enhanced training of health care and medical professionals at both primary and secondary care level to diagnose and treat fragility fractures early in the most vulnerable dementia patients in order to improve patient care and reduce NHS burden by preventing hip fractures [50,51]. A specifically designed quality improvement collaborative work among different community and secondary care teams to identify existing fragility fractures could also improve secondary fracture prevention care [52].

## 5. Conclusions

Dementia is an important risk factor for falls and hip fracture. Dementia patients with a prevalent fragility fracture are associated with a statistically significant higher risk of a new care home placement following an acute hospital admission. This study also showed that dementia patients with prevalent fragility are also at a significant higher risk of a new hip fracture in the next three years. Therefore, an integrated approach between the memory service and bone health clinics and an aggressive osteoporosis treatment may reduce fragility fracture incidence in this high-risk group. Whilst clinical judgement remains crucial in the care of frail older people, it is prudent to consider medical management of osteoporosis in dementia, but only if deemed to be beneficial following the comprehensive geriatric assessment.

## Figures and Tables

**Figure 1 geriatrics-05-00099-f001:**
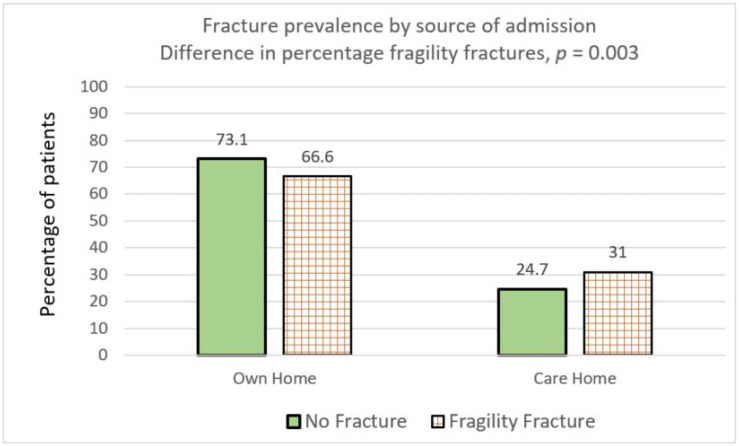
Prevalence of fragility fracture in the community and care home in patients with dementia.

**Figure 2 geriatrics-05-00099-f002:**
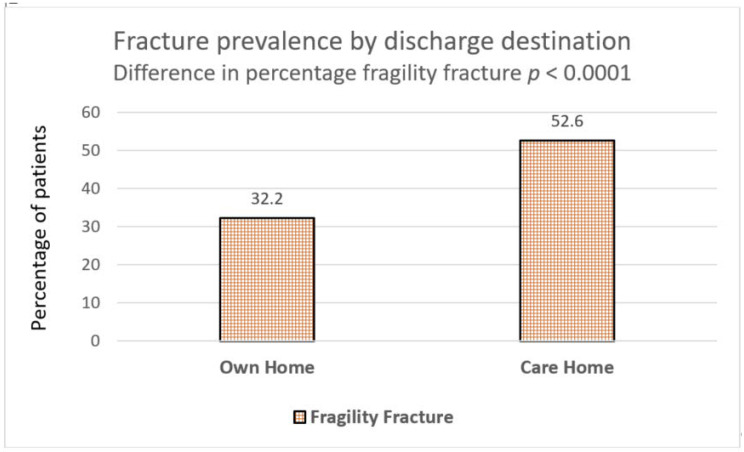
Prevalence of fragility fracture as defined by discharge destination following admission to hospital in acute dementia patients.

**Figure 3 geriatrics-05-00099-f003:**
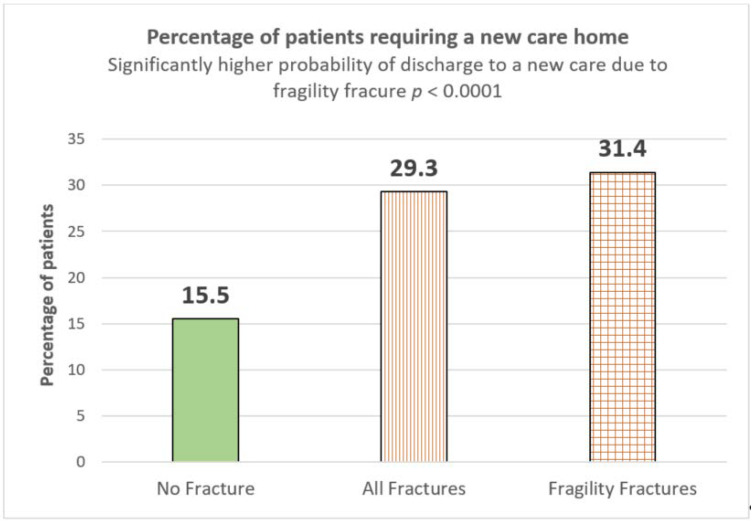
Patients requiring a new care home following an acute admission to hospital.

**Figure 4 geriatrics-05-00099-f004:**
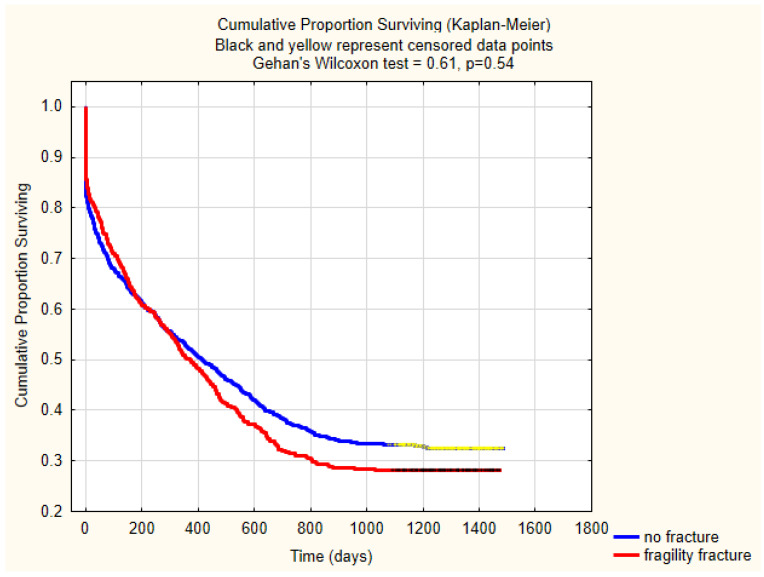
Kaplan–Meier graph showing mortality trends between no fracture and fragility fracture group.

**Table 1 geriatrics-05-00099-t001:** Profile of dementia and prevalent fracture.

Dementia % (*n*)		Group 1: No Fracture	Group 2: All Fracture	Group 3: Fragility Fracture	*p*-Value Group 1 vs. Group 2	*p*-Value Group 1 vs. Group 3
		65.9% (1363/2067)	34.1% (704/2067)	28.7% (595/2067)		
Mean age (SD)		83.8 (8.0)	85.9 (6.9)	85.8 (6.9)	<0.0001	<0.0001
Female % (*n*)		55.2% (752/1363)	76.1% (536/704)	76.7% (457/595)	<0.0001	<0.0001
Living in % (*n*)	Own Home	73.1% (997/1363)	66.8% (470/704)	66.6% (396/595)	0.0028	0.0035
	Care Home	24.7% (337/1363)	30.9% (218/704)	31.0% (185/595)	0.0026	0.0037
Mean CCI (SD)		5.92 (1.47)	6.9 (1.42)	6.02 (1.42)	0.17	0.16
Mean Drugs (SD)		7.76 (3.8)	8.24 (3.5)	8.32 (3.7)	0.005	0.002
Mean Anti-psychotics % (*n*)		16.4% (220/1341)	17.1% (120/700)	16.2% (96/592)	0.69	0.091

*n*: total number of patents; SD: standard deviation.

**Table 2 geriatrics-05-00099-t002:** Clinical outcomes and hip fracture data.

Dementia Patients % (*n*)		Group 1: No Fracture % (*n*)	Group 2: All Fracture	Group 3: Fragility Fracture	*p*-Value Group 1 vs. Group 2	*p*-Value Group 1 vs. Group 3
		65.9% (1363/2067)	34.1% (704/2067)	28.7% (595/2067)		
Mean LoS		17.37 (23.4)	18.8 (23.2)	18.6 (2.37)	0.20	0.35
Discharge destination	Own home	46.3% (632/1363)	32.5% (229/704)	30.7% (183/595)	<0.0001	<0.0001
	Care home	32.2% (439/1363)	50.4% (355/704)	52.6% (313/595)	<0.0001	<0.0001
% Requiring new care home		15.5% (207/1333)	29.3% (201/678)	31.4% (182/580)	<0.0001	<0.0001
Mortality	Inpatient	16.7% (228/1363)	13.6% (96/704)	13.3% (79/595)	0.066	0.057
	30-days	23.5% (320/1363)	19.2% (135/704)	19.4% (115/595)	0.025	0.045
	One-year	47.7% (650/1363)	50.4% (355/704)	49.9% (296/595)	0.2454	0.37
	Three-year	73.5% (352/479)		65.1% (125/192)		0.03
Inpatient hip fracture		2.1% (18/1363)	3.1% (42/354)	3.7% (42/595)	0.26	0.040
Post-discharge hip fracture % (*n*)		2.9% (34/1135)	19.9% (121/608)	21.5 (111/516)	<0.0001	<0.0001

*n*: total number of patents; SD: standard deviation.

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
