# Peer review of "Relationship of Prevalent Fragility Fracture in Dementia Patients: Three Years Follow up Study"

_geriatrics, 2020, doi:10.3390/geriatrics5040099_

Round 1
Reviewer 1 Report
My suggestions:
I suggest to add some graph or figure, which compares the differences between each groups (no fracture, all fracture and fracture fragility) in terms of risk for dementia.
I would suggest to discuss on the relationsip between dementia associated pathways (amyloid metabolism) and calcium metabolism. Also, in introduction I would like to discuss more on calcium metabolism and bone fracture. There may be link between dementia and bone fragility through the impairment of calcium metabolism. Authors should consider this.
Author Response
Dear reviewer, we are very grateful to you for your comments.
We have added 4 new graphs and figures and have gone through our results again to make it easier to understand. We could see how much difference it has made after your review and we have revised results and we have also added 3 year moratlity analyses now. KM survival curve has also been added to make it easy to understand.
We have also included a new paragraph for calcium / vitamin D metabolism in introduce dysregulation of neuronal calcium in relation to Alzhemer’s disease.
Thanks aagin for your time and comments.
Reviewer 2 Report
Dear Authors,
The authors conducted the retrospective observational cohort study to analyze the cause and effect relationship between dementia and fragility fractures. The dementia patients were grouped into three to assess the clinical outcome for three years regarding the length of hospital stay, need of a new care home, post-discharge hip fracture data, and mortality. The study revealed that mean age was significantly higher in dementia patients with a previous fracture and the number of females was significantly higher than that of males. The significantly higher number of dementia patients with previous fractures were living in care homes and taking a significantly higher number of medications. The significantly higher number of dementia patients with previous fragility fracture sustained a new hip fracture, compared to those without previous fracture. The authors concluded that dementia patients with a previous fracture have a significantly high risk of worse clinical outcome and a new hip fracture, proposing that the need of osteoporosis treatment, especially for dementia patients with previous hip fracture.
Please consider the following parts:
- Page1, Line 24:” three groups”, Please present three groups clearly.” no previous fractures; any fractures; and fragility fractures” or “no previous fractures, any fractures, and fragility fractures”?
- Page 2, Line 56: Please cite a reference.
- Page 2, Line 58: Please cite a reference.
- Page 2, Line 60: Please cite a reference.
- Page 3, Line 83: ” three groups”, Please present three groups clearly.” no previous fractures; any fractures; and fragility fractures” or “no previous fractures, any fractures, and fragility fractures”?
- Page 4, Line 129: “number of medications”, How is it counted? What kind of drugs were prescribed? It may deserve to present class of drugs in Supplement.
- Page 4, Line 134: The reviewer recommends the use of graphs for statistically significant results. A graph is more visual and helps readers grasp the significance of the manuscript. The table can be placed in Appendix or Supplementary Data.
- Page 5, Line 156: The reviewer recommends the use of graphs for statistically significant results.
- Page 6, Discussion: Although Mean Charlson comorbidity index did not show any significance among the groups, the reviewer recommends analyzing and discussing disease types which may be associated with fragility, osteoporosis, fall, and inflammation, among others and residential care facility.
Suggested references:
Tomaszewska, E.; Muszyński, S.; Kuc, D.; Dobrowolski, P.; Lamorski, K.; Smolińska, K.; Donaldson, J.; Świetlicka, I.; Mielnik-Błaszczak, M.; Paluszkiewicz, P.; Parada-Turska, J. Chronic dietary supplementation with kynurenic acid, a neuroactive metabolite of tryptophan, decreased body weight without negative influence on densitometry and mandibular bone biomechanical endurance in young rats. PLoS One 2019, 14(12), e0226205.
Pérez-Pérez, A.; Sánchez-Jiménez, F.; Vilariño-García, T.; Sánchez-Margalet, V. Role of Leptin in Inflammation and Vice Versa. Int. J. Mol. Sci. 2020, 21, 5887.
López-Gambero, A.J.; Sanjuan, C.; Serrano-Castro, P.J.; Suárez, J.; Rodríguez de Fonseca, F. The Biomedical Uses of Inositols: A Nutraceutical Approach to Metabolic Dysfunction in Aging and Neurodegenerative Diseases. Biomedicines 2020, 8, 295.
Kim, E.Y.; Ahn, H.-S.; Lee, M.Y.; Yu, J.; Yeom, J.; Jeong, H.; Min, H.; Lee, H.J.; Kim, K.; Ahn, Y.M. An Exploratory Pilot Study with Plasma Protein Signatures Associated with Response of Patients with Depression to Antidepressant Treatment for 10 Weeks. Biomedicines 2020, 8, 455.
Landi, G.; Marchi, M.; Ettalibi, M.Y.; Mattei, G.; Pingani, L.; Sacchi, V.; Galeazzi, G.M. Affective and Sexual Needs of Residents in Psychiatric Facilities: A Qualitative Approach. Behav. Sci. 2020, 10, 125.
Tanaka, M.; Bohár, Z.; Vécsei, L. Are Kynurenines Accomplices or Principal Villains in Dementia? Maintenance of Kynurenine Metabolism. Molecules 2020, 25, 564.
Tanaka, M.; Toldi, J.; Vécsei, L. Exploring the Etiological Links behind Neurodegenerative Diseases: Inflammatory Cytokines and Bioactive Kynurenines. Int. J. Mol. Sci. 2020, 21, 2431.
Park, S.; Bak, A.; Kim, S.; Nam, Y.; Kim, H.; Yoo, D.-H.; Moon, M. Animal-Assisted and Pet-Robot Interventions for Ameliorating Behavioral and Psychological Symptoms of Dementia: A Systematic Review and Meta-Analysis. Biomedicines 2020, 8, 150.
Landi, G.; Marchi, M.; Ettalibi, M.Y.; Mattei, G.; Pingani, L.; Sacchi, V.; Galeazzi, G.M. Affective and Sexual Needs of Residents in Psychiatric Facilities: A Qualitative Approach. Behav. Sci. 2020, 10, 125.
- Page 8: A list of abbreviation is missing.
- Page 8, References: The number of references is expected to be at least more than 50.
The manuscript contains no figures, two table, and 20 references. The reviewer recommends including more references for original articles, preferably more than 50. The limitation and strength of the study were accurately described. Typological errors are noticeable in the manuscript and the reference style should be corrected for Geriatrics.
The manuscript is of great value analyzing the cause and effect relationship between dementia and fragility fractures and suggesting a personalized treatment to improve the quality of life of dementia patients after hospital discharge.
I declare no conflict of interest regarding this manuscript.
Author Response
Dear Reviewer
Many thanks for taking time to share your comments and we are very grateful to you for your time.
We have gone through each and every point and please find our response as below.
- We have presented three groups clearly both in the abstract and methods.
- New refernces has been added as below in the introduction
- Wang, H.K.; Hung, C.M.; Lin, S.H. et al. Increased risk of hip fractures in patients with dementia: a nationwide population-based study. BMC Neurol, 2014, 14, 175.
- Rapp, K. People with Alzheimer's disease are at increased risk of hip fracture and of mortality after hip fracture. Evidence-Based Nursing 2011, 14, 78-79.
- Bai, J.; Zhang, P.; Liang, X.; Wu, Z.; Wang, J.; Liang, Y. Association between dementia and mortality in the elderly patients undergoing hip fracture surgery: a meta-analysis. J Orthop Surg Res, 2018,13, (1): 298.
- Berry, S.D.; Rothbaum, R.R.; Kiel, D.P.; Lee, Y.; Mitchell, S.L. Association of clinical outcomes with surgical repair of hip fracture vs nonsurgical management in nursing home residents with advanced dementia. JAMA Intern Med, 2018, 178 (6):774-780.
- Menzies, I.B.; Mendelson, D.A.; Kates, S.; Friedman, S.M. Prevention and clinical management of hip fractures in patients with dementia. Geriatr Orthop Surg Rehabil, 2010, 1 (2): 63-72.
- Morrison, R.S.; Siu, A.L. Survival in end-stage dementia following acute illness. JAMA, 2000, 284, 1, 47-52.
- Waran, E.; William, L. Hip fractures and dementia: clinical decisions for the future. Oxf Med Case Reports, 2016, (2): 19-21.
- We only counted the total number of drugs and class of the drugs was not recorded and this has been one of the limitation of the study as detailed association was not studied for anti-resorptive or dementia drugs.
- 4 new graphs / figures have been added to explain our results. KM survival curve has also been added for 3 year mortality analyses.
- In this study, Mean Charlson comorbidity index did not show any significance among the groups but we have not studied the relation of fragility fracture with co-morbidties. This has been accepted as a limitation of the study.
Please do not hesitate to write back if you have any further comments.
Thanks again for your time and support.
Round 2
Reviewer 1 Report
Authors fulfilled my suggestions.
Author Response
Thans for your support.
Reviewer 2 Report
Dear Authors,
The manuscript presented that dementia patients with a previous fracture have a significantly high risk of worse clinical outcome and a new hip fracture, proposing that the need of osteoporosis treatment, especially for dementia patients with previous hip fracture.
Please reconsider the following parts:
- Pages 4, 6: The reviewer recommends coloring the graphs. Graphs in color is generally more visual and help readers grasp the significance of the manuscript.
- Page 10, References: The number of references is still low. Here is a list of suggested references:
1.Tomaszewska, E.; Muszyński, S.; Kuc, D.; Dobrowolski, P.; Lamorski, K.; Smolińska, K.; Donaldson, J.; Świetlicka, I.; Mielnik-Błaszczak, M.; Paluszkiewicz, P.; Parada-Turska, J. Chronic dietary supplementation with kynurenic acid, a neuroactive metabolite of tryptophan, decreased body weight without negative influence on densitometry and mandibular bone biomechanical endurance in young rats. PLoS One 2019, 14(12), e0226205.
2. Pérez-Pérez, A.; Sánchez-Jiménez, F.; Vilariño-García, T.; Sánchez-Margalet, V. Role of Leptin in Inflammation and Vice Versa. Int. J. Mol. Sci. 2020, 21, 5887.
3. López-Gambero, A.J.; Sanjuan, C.; Serrano-Castro, P.J.; Suárez, J.; Rodríguez de Fonseca, F. The Biomedical Uses of Inositols: A Nutraceutical Approach to Metabolic Dysfunction in Aging and Neurodegenerative Diseases. Biomedicines 2020, 8, 295.
4. Tanaka, M.; Toldi, J.; Vécsei, L. Exploring the Etiological Links behind Neurodegenerative Diseases: Inflammatory Cytokines and Bioactive Kynurenines. Int. J. Mol. Sci. 2020, 21, 2431.
5. O’Day, D.H. Calmodulin Binding Proteins and Alzheimer’s Disease: Biomarkers, Regulatory Enzymes and Receptors That Are Regulated by Calmodulin. Int. J. Mol. Sci. 2020, 21, 7344.
- Page 9, Conclusion: This section is too short. The reviewer recommends including more descriptions such as future perspectives.
- Page 10: A list of abbreviation is missing.
The manuscript contains four figures, two table, and 38 references. The reviewer recommends including more references for original articles, at least 50. The limitation and strength of the study were accurately described. The reference style should be corrected for Geriatrics. The manuscript has been improved, but some reviewer’s comments were ignored and thus the authors are expected to reconsider them. The manuscript is of great value analyzing the cause and effect relationship between dementia and fragility fractures and suggesting a personalized treatment to improve the quality of life of dementia patients after hospital discharge.
I declare no conflict of interest regarding this manuscript.
Author Response
Dear Reviewer
Many thanks for taking time to share your comments.
We have revised the whole version again.
We have revised all the graphs and made it colour
Whole manuscript has been revised for abreviations and duplictaion
As suggested by you as a requirement to be considered as an original article, we have included more references and we found references suggested by you very helpful and we have included 3 references.
We have also revised the conclusions
Please do let us know, if ay further chages needed, it will help our manuscript to improve further.
Thanks
Authors